gender inequality; development; parental relationship; adolescent mental health

**Corresponding author:**
Nicolas Crossley;
Email: ncrossley@uc.cl

# Parental gender inequality and their children's educational attainment, quality of life and mental health: An analysis from the Pelotas 1993 birth cohort in Brazil

Nicolas A. Crossley[1,2,3,4] , Leticia Czepielewski[5], Ana M.B. Menezes[6], Fernando Wehrmeister[6] and Clarissa S. Gama[7]

[1]Department of Psychiatry, School of Medicine, Pontificia Universidad Catolica de Chile, Chile; [2]Centro de Interés Nacional para Investigación e Innovación en Niñez, Adolescencia, Resiliencia y Adversidad, IINARA, Chile; [3]Department of Psychiatry, Antioquia University, Colombia; [4]Department of Psychiatry, University of Oxford, United Kingdom; [5]Departamento de Psicologia do Desenvolvimento e da Personalidade, Programa de Pos-Graduacao em Psicologia, Universidade Federal do Rio Grande do Sul, Brazil; [6]Universidade Federal de Pelotas, Brazil and [7]Department of Psychiatry, Universidade Federal do Rio Grande do Sul, Brazil

## Abstract

Gender, as a sociostructural factor, may shape child development through social norms that influence family dynamics. We examined whether more egalitarian parental relationships are associated with better developmental outcomes. Using data from the Pelotas 1993 birth cohort (Brazil), we adapted a population-level gender inequality metric to characterise parental relationships. The Couple's Gender Inequality Index (CGII) was derived from maternal health, parental education and income. Associations between CGII and educational attainment, quality of life, and depression at age 18 were assessed using linear regression models adjusted for family income, gestational age, birth weight, parental cohabitation and race. The sample comprised 2,852 participants (1,446 women). Higher CGII scores, indicating greater equality within couples, were associated with significantly higher educational attainment in both females and males. Higher quality of life at age 18 was observed in the second and fourth CGII quartiles compared with the most unequal. Greater equality was associated with lower risk of depression at age 18, although this association was not robust to adjustment. Among girls, a similar pattern was observed for emotional symptoms at age 15. Overall, greater couple-level gender inequality was associated with poorer developmental outcomes in offspring.

## Impact statement

Gender norms shape children's lives. However, most measures of gender inequality are designed for comparing countries or large communities. While these indicators are valuable, they cannot capture how gender dynamics influence the development of an individual child. Crossley *et al.* address this gap by developing the CGII, a tool that reflects the balance of resources, opportunities and health between mothers and fathers in a particular household. By applying the CGII to a long-running birth cohort in southern Brazil, they show that this family-level measure of gender equality is meaningfully linked to children's later outcomes. Young people who grew up in more gender-equal households tended to achieve higher levels of education and reported a better quality of life in adolescence. They also showed fewer symptoms of depression and emotional difficulties. These associations were seen for boys and girls, suggesting that gender equality within the family benefits all children. The broader impact of this work is the demonstration that gender inequality can be measured in a way that is practical, scalable and relevant for both research and policy. The CGII allows for the identification of families who may be experiencing unequal gender dynamics, in contrast to national or regional indicators that mask variation within communities. This tool can support more targeted interventions, help evaluate programmes aimed at improving parental equity and deepen our understanding of how everyday family structures contribute to long-term developmental and mental health outcomes.

## Introduction

Gender is a complex system of social difference and inequality that goes beyond an individual's attribute (Homan, 2019). It can have a major impact on people's lives and health (Heise et al., 2019), even from an early age. Gender norms are acquired during early adolescence (Blum et al.,

2017), and evidence suggests that they modulate the risk of developing mental health problems such as depression in both young people (Koenig et al., 2021) and adults (Seedat et al., 2009). Furthermore, social factors have been proposed to become biologically "embedded" during development, with long-term consequences (Hertzman, 2012), a process observed in the brains of adult women living in gender unequal environments (Zugman et al., 2023). There is a growing call to incorporate gender-related factors into research (Barr et al., 2024), particularly in studies of young people during their developmental years.

A recognised challenge in the field is that methods to quantify the effects of gender inequality on health remain underdeveloped (Weber et al., 2019). Arguably, the best-known metrics are country-level multi-dimensional indices that summarise the disadvantages faced by women across the world, such as the United Nation's Gender Inequality Index (Gaye et al., 2010). This country-level perspective has been valuable in Global Development research and has helped to examine associations between gender inequality and mental health problems (Yu, 2018) or educational attainment (Guiso et al., 2008) at the population level. Gender inequality can also be measured at smaller scales, allowing differences between communities within the same country to be explored (Ewerling et al., 2017). Moreover, gender inequality could be measured at the family level, examining how gender norms and structures shape relationships between parents and their children (Brines, 1994). Families have a significant impact on children's social and emotional development (Grusec, 2011), providing resilience to environmental adversities such as poverty. A previous study found increased emotional and behavioural difficulties among children whose mothers reported perceived gender discrimination (Stepanikova et al., 2022). To better understand the associations between gender inequality and child development, there is a need for new metrics beyond national or community-level characteristics that can be applied at the individual level.

In this study, we analysed educational outcomes, quality of life and mental health in adolescents from the Pelotas 1993 birth cohort (Gonçalves et al., 2018), according to the degree of gender inequality within the families in which participants were raised. The setting is nontrivial: Latin America has one of the highest rates of gender inequality in the world (Camou and Maubrigades, 2017). To measure gender inequality at the family level, we adapted the dimensions of the UN Gender Inequality Index (Gaye et al., 2010) to the interspousal relationship (Homan, 2019). Our metric was designed to shed light on how the structure of the relationship between parents that may have enduring consequences for the children. Specifically, the CGII was developed to mirror the country-level Gender Inequality Index from the United Nations, which captures gender-based disadvantage across three dimensions: reproductive health (perinatal care and adolescent pregnancy of the mother), empowerment (educational level of the mother compared to the father) and labour market participation (comparison of the individual earnings from both parents). Poor reproductive health, lower educational level and lower earnings for the mother compared with the father were considered indicators of reduced power and opportunity for women, reflecting greater gender inequality and resulting in a lower CGII value.

Considering the known impact of adverse environments on women, we hypothesised that girls (but not boys) growing up in gender-unequal families would present lower educational attainment and quality of life, and higher rates of depression.

## Methods

### Design

This study forms part of a population-based birth cohort. A STROBE checklist is provided in the Supplementary Information.

The Pelotas 1993 birth cohort (Victora et al., 2008) includes 5,265 births from women living in the city of Pelotas, Brazil . At cohort intake, 18.4% belonged to families receiving less than the minimum wage, 9.8% presented birth weight < 2,500 g, and 11.2% were born before 37 weeks (Gonçalves et al., 2018). The cohort has been followed up at ages 11, 15, 18 and 22 years, with respective follow-up rates of 87.5%, 85.7%. 81.4% and 76.3% (Gonçalves et al., 2018). We used information collected at birth, and at 15 and 18 years.

### Metrics

#### Couple's gender inequality index

We followed previous approaches examining disparities in power and resource allocation between men and women within marriages (Homan, 2019), which may significantly affect the development of children through the inter-generational transmission of gender norms, resources and support. We adapted the multi-dimensional approach used in the creation of the UN Gender Inequality Index (Gaye et al., 2010) to measure disparities within the parental relationship, using data available from the Pelotas 1993 Birth cohort as shown in Table 1.

Reproductive health is not directly anchored to a male-based outcome, unlike empowerment or labour market participation, and therefore may reflect factors affecting both genders, such as socioeconomic status (Permanyer, 2013). However, female empowerment has been shown to influence reproductive health indicators in previous studies (Bagade et al., 2022).

Each dimension was scored on a 0–1 scale, with 0 indicating maximum inequality and 1 indicating parity. In the case of reproductive health, which had two indicators, scores were averaged. The resulting CGII is a single indicator calculated as the mean of the three-dimension scores for each couple.

**Table 1.** Dimension of the UN Gender Inequality Index and the proposed Couple's Gender Inequality Index

| Dimension | Population-based indicators used in the UN Gender Inequality Index | Parental characteristics used in the Couple's Gender Inequality Index |
|---|---|---|
| *Reproductive health* | - Maternal perinatal mortality (as an indicator of quality of perinatal care)<br>- Adolescent pregnancy. | - Number of perinatal visits during the index pregnancy (as a measure of quality of perinatal care).<br>- Age of mothers at first childbirth. |
| *Empowerment* | - Women's educational attainment compared with men's (secondary or higher)<br>- Presence of female seats in parliament. | - Ratio between the mother's and father's educational attainment (years of education) at the child's birth. |
| *Labour market* | - Female participation rate compared with men's. | - Ratio between the mother's and father's income. |

Specific scoring details were as follows:

- *Number of perinatal visits*: a value of 0 if the index birth was not preceded by any perinatal visits of the mother, and 1 if the number of visits was equal or above 8 (as recommended by the World Health Organization). The number of visits between 1 and 7 was proportionally given a value between 0 and 1.
- *Age when mothers gave birth to their first child*: first deliveries aged 20 or above were scored as 1. Deliveries at 14 years or less (early adolescent pregnancies per World Health Organization definition) as 0, and ages 15–19 were proportionally assigned values between 0 and 1.
- *Ratio of maternal to paternal education*: the ratio was capped at 1 when mothers had more years of education than the father. A supplementary analysis explored the cases when paternal education was lower than maternal.
- *Ratio of maternal to paternal income*: based on the average of data collected at birth and 15 years. Averaging was used because pregnancy may temporarily lead women to leave the labour market. At birth, income contribution was categorised as follows: father sole breadwinner (0), both parents contributing (0.5), or mother as main breadwinner (1). At 15 years, absolute income for each parent was used, and ratios were capped at 1 when the mother earned more than the father.

We capped the ratio of maternal to paternal education or income at 1 when the mother had more education or income than the father. This approach aligns with the UN Gender Inequality that it mirrors, where the focus is on contexts in which systemic disadvantage overwhelmingly affects women, rather than on comparing female and male advantages or disadvantages. We also report exploratory analyses of an index focusing on the father's disadvantage relative to the mother, particularly in the education dimension, where fathers were most disadvantaged, and its association with developmental outcomes. In this case, the education index represents the ratio between the father's and mother's, capped at 1 if the father has equal or higher education than the mother.

Cronbach's alpha was used to explore the internal consistency of the index. Associations for each individual dimension are presented in the Supplementary Information.

### Outcomes

We examined the associations between gender inequality and several developmental outcomes from childhood to young adulthood. First, we included a measure of educational attainment at 18 years, specifically the number of completed years in education obtained from self-report from the adolescent. Education has a significant impact on people's future, including their health and lifetime earnings. In Brazil, education is compulsory from ages 6 to 14 (9 years), followed by 3 years of non-compulsory upper secondary education (ages 15 to 17). Second, we examined quality of life at 18 years, assessed using the overall score of the WHO Quality of Life-brief (QoL) (Fleck et al., 2000), with higher scores indicating better quality of life. Finally, we looked at the emergence of mental health problems, particularly depression symptoms, at two points. The focus on depression is based on previous evidence suggesting a modulatory role of gender norms (Seedat et al., 2009; Koenig et al., 2021). At 15 years, emotional symptoms were assessed using the emotion subscale of the Brazilian version of the Strengths and Difficulties Questionnaire (SDQ) (Fleitlich-Bilyk and Goodman, 2004), using the responses from the main caregiver as in a previous report from this cohort (Anselmi et al., 2012). The emotion subscale has been related to the risk of depression (Armitage et al., 2023),

providing an early indicator of emerging emotional problems. At age 18, we looked at the association with a diagnosis of depression at that age (binary outcome) using the Mini International Neuropsychiatric Interview (MINI).

### Other confounders
We also included several confounders in our fully adjusted model:

○ total family income at birth (parental or other sources).
○ one or more parents describing themselves as "non-white" at birth.
○ gestational age (in weeks) and birth weight.
○ living with both parents at 18 years old.

Participants were included if data were available to calculate the CGII. Characteristics of excluded participants (both with incomplete data or lost to follow-up) are reported in the Supplementary Information.

### Statistical analyses

We used generalised linear models to examine the associations between the CGII and educational attainment, quality of life and mental health at 15 and 18 years. To offer a nuanced interpretation of the Couple's Gender Inequality's association without forcing a strict linear or multiplicative relationship, we examined models according to bins of quartiles of the CGII. We applied regression models appropriate to each outcome's distribution. For count outcomes like years of education and SDQ emotional symptoms score, we used the Poisson regression model with a log link to estimate relative differences (rate ratios) in years of education across quartiles of gender inequality. The coefficients were exponentiated to express multiplicative effects. For the binary depression outcome, we used logistic regression with a logit link for binary outcomes, with exponentiated coefficients corresponding to odds ratios. For quality of life (approximately continuous), linear regression was used. The general form of the models used was the following:

$$y_i = \alpha_i + \beta_1 CGII_i + \beta_2 Sex_i + \beta_3 (CGII_i \times Sex_i) + \epsilon_i$$

With a fully adjusted model adding the following confounders: family income at birth, one or more parents describing themselves as non-white, gestational age and birth weight, and whether they were living with both parents at age 18. Stepwise models showing the individual and incremental weight of these confounders are included in the Supplementary Information.

Considering the importance of socioeconomic factors in vulnerable populations and the importance of considering interacting vulnerabilities during development, we examined interactions between CGII and family income.

### Results

### Characteristics of the sample

A total of 2,852 participants (1,446 female) born in 1993 had data that allowed us to construct a CGII from the assessments performed and were included in the analyses. They represented 54% of the population born in 1993 in Pelotas, and 69.5% of those successfully followed up at age 18. Participants not included were from families with lower incomes, their mothers had fewer years of education, received worse perinatal care and were more likely to have been adolescents (Supplementary Table S1).

**Table 2.** Characteristics of the parents used to build the proposed Couple's Gender Inequality Index

| Dimension | Number for defined subgroup | | |
|---|---|---|---|
| Perinatal care N(%) | Insufficient (less than 4) | Intermediate care | Good (8 or more perinatal visits) |
| | 217 (7.6%) | 1021 (35.8%) | 1614 (56.6%) |
| Mother's age at birth N(%) | ≤14 years | 15–19 years | > = 20 years |
| | 29 (1.0%) | 834 (29.2%) | 1989 (69.7%) |
| Parental education N(%) | | Higher education father | Equal or higher education mother |
| | | 1,058 (37.1%) | 1,794 (62.9%) |
| Parental earnings N(%) | | Higher earnings father | Equal or higher earnings mother |
| | | 2,711 (95.1%) | 141 (4.9%) |

Table 2 describes the characteristics of the parents of participants used to construct the Couple's Gender Inequality. Slightly more than half of mothers of those included had good perinatal care (≥8 perinatal visits), while the remainder had less than the recommended number, with 7.6% receiving poor according to World Health Organization criteria. Almost one third of mothers gave birth to their first child during adolescence, and 1% had a very young pregnancy (14 years old or less). In terms of education, 62.9% of the couples had equal level of education or mothers with more years of education than fathers. In contrast, only 4.9% had equal levels of income or mothers earning more than fathers. The distribution of the CGII and its different dimensions is shown in Figure 1. Cronbach's alpha was low (α = 0.24), suggesting that the components captured distinct dimensions of inequality rather than a single underlying construct. Associations between each dimension and the outcomes, analysed independently, are presented later in the "Results" section.

### Associations with educational attainment

The mean number of years of education was 8.9, with 4.9% of the sample attaining a maximum of 12 years, and 3.7% of 4 years or less. The CGII was positively associated with an increased number of attained years (Figure 2A, Supplementary Table S2). Compared with children from families in the lowest quartile of the CGII (most unequal), those in the second quartile had higher levels by an 10.2% [4.3, 16.4%], 14.6% [8.6, 20.9%] in the third quartile, and 23.3% [16.9, 30.1%] in the highest quartile in the unadjusted model. Corresponding estimates in the fully adjusted model were 8.5% [2.7, 14.7%], 12.8% [6.8, 19.2%] and 20.4% [14.0, 27.2%]. There were no significant interactions between the CGII and sex in the unadjusted model, but a significant interaction emerged in the fully adjusted model between the highest CGII quartile and sex (P = 0.04), showing a negative interaction for girls (*i.e.* smaller increases in education with higher equality; see Supplementary Table S2). Considering that 12 was the maximum number of years attainable

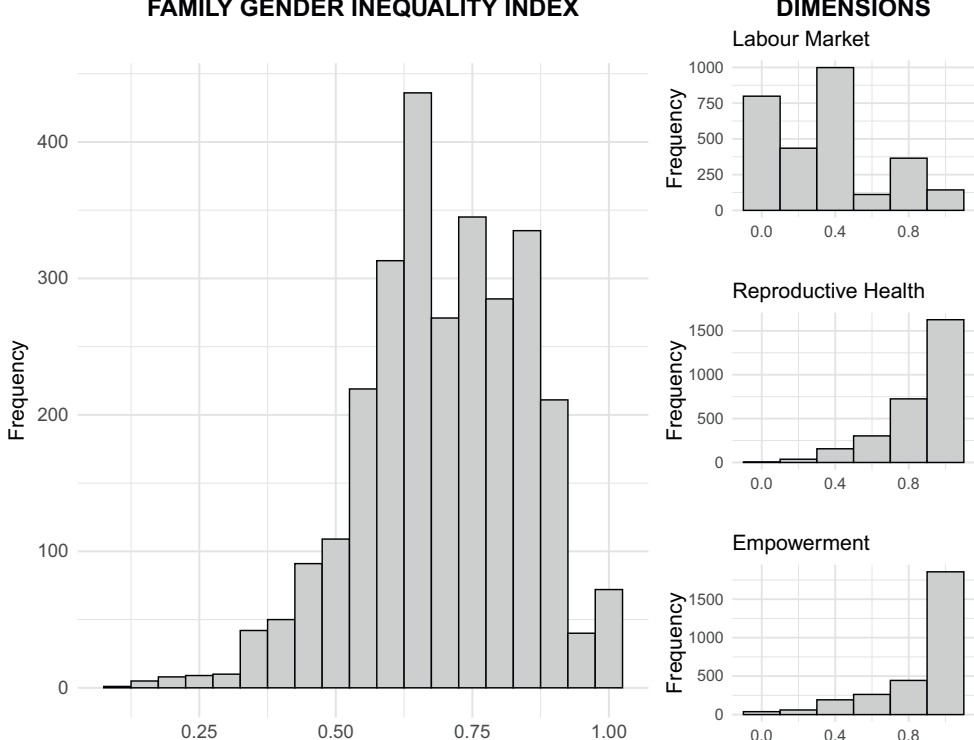

**Figure 1.** Distribution of Couple's gender Inequality Index and its dimensions.

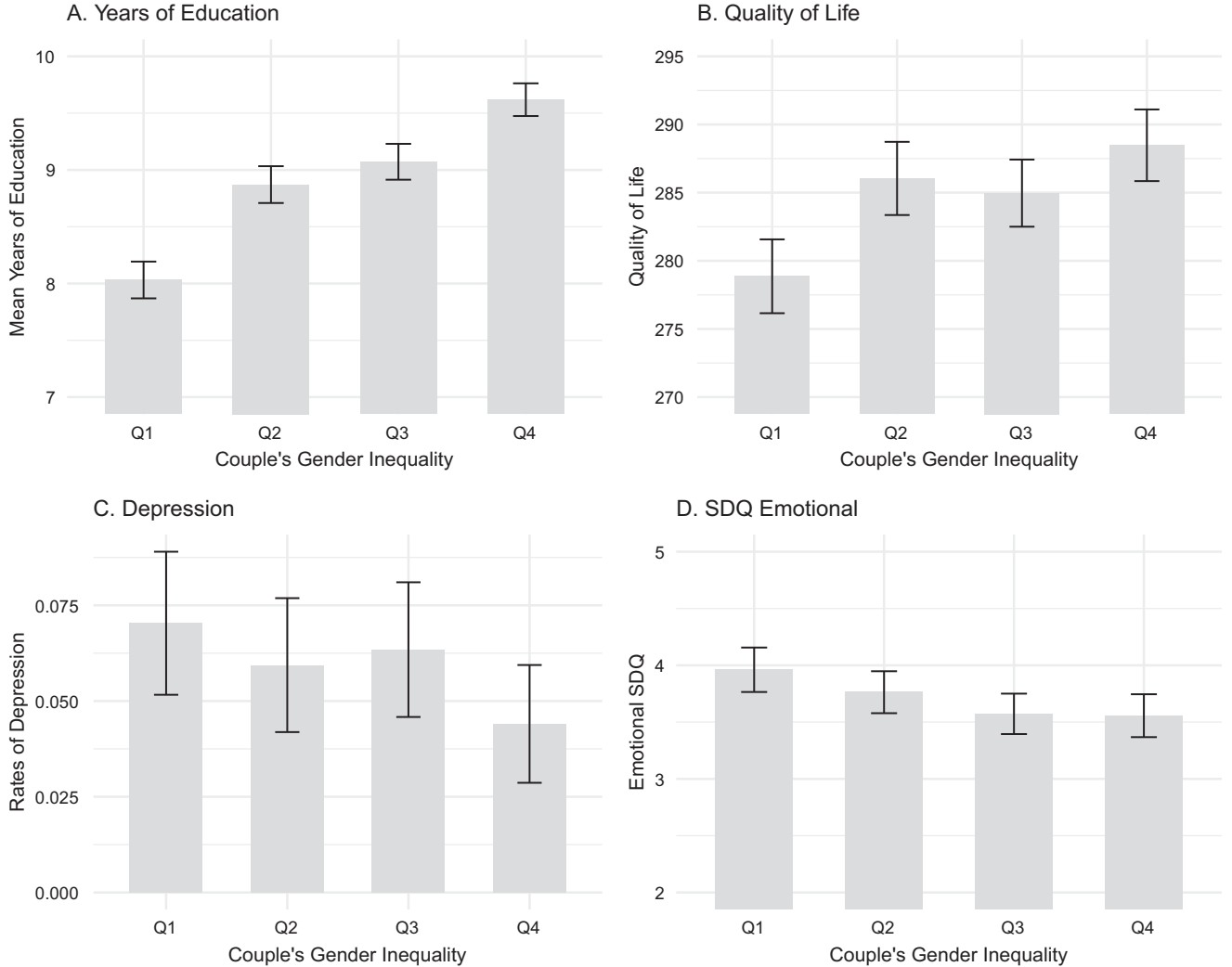

**Figure 2.** Associations between the Couple's Gender Inequality Index and Education (A), quality of life (B), risk to depression at age 18 (C) and SDQ emotional scores at age 15. *Values are shown per quartile of Couple's gender Inequality without correction for other confounders (sex, family income, gestational age, birth weight or non-white parent).*

at that age, this negative interaction likely reflected a ceiling effect among girls, who already had higher educational levels than boys (Supplementary Figure S1A).

We also found a significant negative interaction between family income and the CGII index on educational attainment when comparing the most equal quartile and least equal, both in the unadjusted ($P = 0.005$) and fully adjusted models ($P = 0.028$). In other words, increases in education associated with high CGII (greater equality) were more pronounced in lower-income families. Similar to the sex interaction with CGII, this result may also partly reflect a ceiling effect (Supplementary Figure S1B).

### Associations with quality of life

Quality of life was also associated with CGII. Children from more balanced couples in the second and fourth quartile reported a better quality of life than those in the lowest quartile, both in the unadjusted model (7.9 points [2.4, 13.4] in the second and 8.9 [3.5, 14.4] in the fourth) and in the fully adjusted model (6.7 points [1.4, 12.2] in the second and 6.6 [1.1, 12.1] in the fourth) (Figure 2B, Supplementary Table S3). We did not observe significant

interactions between the CGII and sex or between the CGII and family income.

### Associations with mental health

At age 18, 5.9% of participants fulfilled criteria for a depressive disorder according to the MINI. There was a significant negative association between the CGII and the odds of depression (Figure 2C). This was significant in the unadjusted model for the second quartile (40.7% [15.6, 94.9%] of the odds of the first quartile) and those in the highest CGII quartile (34.9% [12.5, 84.5%]), but not in the fully adjusted model (44.6% [17.4, 106.4%] for the second quartile, and 43.4% [15.4, 107.0%] for the highest quartile) (Supplementary Table S4). We found no statistically significant interactions between CGII and sex or family income.

At age 15, Emotional SDQ scores were lower in children from families with higher CGII (Figure 2D), although this was not statistically significant in either the unadjusted or fully adjusted model. There was a significant negative interaction between CGII and sex in the second quartile both in the unadjusted ($P = 0.0002$) and fully adjusted models ($P = 0.001$) (Supplementary Table S5). As shown

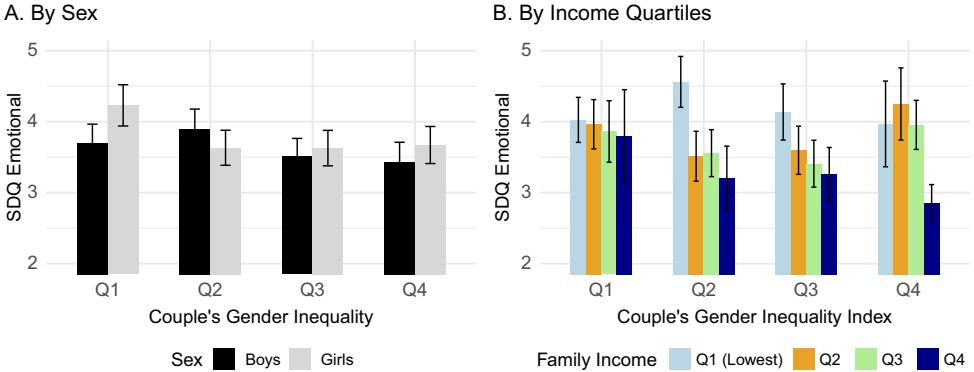

**Figure 3.** Unadjusted SDQ Emotional scores and their association with Couple's Gender Inequality according to (A) sex and (B) family Income.

in Figure 3A, girls in the second quartile (but not boys) had lower emotional SDQ scores than those from the lowest CGII quartile. The relationship between CGII and income was more complex (Figure 3B), with children in the top income quartile benefiting from increased equality in the couple. When examining different quartiles to the first quartile, there was a significant negative interaction between income and CGII in the second quartile compared to the first, where this relationship was stronger.

### Further analyses of index components

We report further analyses looking at all the associations between outcomes and specific domains of the Couple's Gender Inequality in the Supplementary Figure 2. Associations with individual domains show overall concordance with the combined index, indicating that all domains contribute to the associations observed with the composite measure.

There were no clear associations related to families in which the father was at an educational disadvantage compared to the mother, as shown in Supplementary Figure 3.

### Discussion

Gender may have a strong effect on children's development, but its examination has been limited by the lack of appropriate metrics. We present an analysis of the Pelotas 1993 birth cohort in Brazil, using an adaptation of the widely used UN Gender Inequality Index applied here to the parental relationship of the child. Our most consistent finding was that a higher CGII, indicating a more gender-equal family, was significantly associated with better educational outcomes in both girls and boys. We also found that a higher CGII was associated with better quality of life and lower risks of depression at 18 years in boys and girls, although these associations were significant only for specific quartiles. Our hypothesis that gender inequality within families would benefit girls more than boys was partly supported by a significant interaction observed in SDQ Emotional scores at 18 years of age.

Views about how gendered systems work have evolved over recent decades. While early feminist theories considered patriarchal systems as primarily disadvantaging women for the benefit of men, newer approaches suggest that they may negatively affect both women and men (Homan, 2019). Several studies have reported that conformity to rigid masculine norms is associated with worse mental health, possibly by inhibiting help seeking and reinforcing maladaptive coping styles, including alcohol use (Seidler et al.,

2016; Silva et al., 2019). Our initial hypothesis predicted a significant interaction between gender and the CGII, whereby girls would benefit more than boys from being raised by more egalitarian couples. Instead, our results showed that higher CGII were associated with better outcomes in both boys and girls. These findings are consistent with previous evidence suggesting that rigid gender norms are detrimental to both boys and girls (Baird et al., 2019; Koenig et al., 2021). The association with academic performance was particularly clear and aligns with findings from China (Chen et al., 2022; Zhang et al., 2024).

There was a significant association between higher CGII and lower risk of depression at age 18. However, this was not significant at age 15 considering the scores of the emotional subset of the SDQ obtained from the main caregiver report, although the direction of the association was similar. At that age, our data suggested that girls in particular were more affected. Previous studies have indicated that rigid gender norms or gender inequality are associated with higher risk of depression in both genders, particularly during adolescence or young adults (Ali et al., 2017; Koenig et al., 2021). Conversely, population-level studies comparing rates between genders have suggested a greater detrimental effect on women than on men (Seedat et al., 2009; Yu, 2018). One possibility is that there is a critical period effect. Some studies suggest that girls might be more affected during the transition to adolescence, when they face a higher number of psychosocial challenges than boys (Petersen et al., 1991). Our results showing this interaction at 15 partially supports this idea. Other work suggests that the cumulative lifetime disadvantages that women experience become more apparent in mental health at later ages (Bracke et al., 2020).

Overall, the CGII provided information on developmental variation that could not be explained by socioeconomic or racial factors. Although based on static characteristics that are not readily modifiable, it may help identify high-risk children and their parents for whom interventions targeting gender norms might be warranted (Heise et al., 2019). Our findings of a significant interaction between the CGII and family income, where associations with good educational outcomes were stronger in families with low income, point to potential avenues for intervention in vulnerable groups. One could question its usefulness compared with direct measures of parents' gender attitudes, if such data were available. However, we would argue that these would provide complementary insights. As discussed in relation to race as a sociostructural factor (Gee and Ford, 2011), individuals' recognition of disadvantage related to race or gender does not necessarily capture the full extent of the structural barriers they face. Furthermore, self-report may be biased by perceived social desirability.

There are several limitations to our study be acknowledged. First, although we report data from a large population-based cohort with substantial efforts to minimise attrition (Gonçalves et al., 2018), complete data to calculate the index were available for 54% of those born in Pelotas 1993. Participants not included were more likely to come from vulnerable groups with higher rates of adolescent pregnancies, worse perinatal care, lower maternal education and lower family incomes. The Cronbach's alpha of the CGII was low, indicating that the index was not unidimensional or homogeneous. This was expected to some extent, as the design followed that of the UN Gender Inequality Index, a composite measure combining different dimensions that are not necessarily correlated (Permanyer, 2013), and that captures a multidimensional state of disadvantage. Countries have shown that equity can be achieved first in some domains before others, and specific interventions may target particular dimensions (Gaye et al., 2010). As with the country-level UN Gender Inequality Index, there are advantages to having a single summary measure of gender-based adversities faced by children, particularly for decision-making purposes. Our sample presented a relatively low rate of depression at age 18 (6%) compared with other studies in Brazil (Hintz et al., 2023), which reduced the power of our analysis, particularly for detecting interactions. Studies focusing on high-risk populations may be better powered to detect such effects. The prospectively ascertained data from birth allowed us to include perinatal variables such as gestational age or number of perinatal visits with high levels of confidence. One could argue that our analytical choice of binning data into CGII quartiles may lose information and impose arbitrary thresholds. However, this should be balanced against alternatives that force specific data distributions or introduce non-linear terms that are harder to interpret, especially when examining interactions. Finally, we note that our results describe associations and do not necessarily imply causation. We also adjusted for several confounders such as baseline income, race or parental cohabitation, but residual confounding (such as parental mental health) may still explain some associations.

In summary, we demonstrate that the CGII, a metric applied at the family level, shows significant associations with key developmental outcomes. Children from more egalitarian families, as indicated by this index, are more likely to spend longer in education, report a better quality of life and show a lower risk of depression in early adulthood. This index may therefore be a valuable tool for exploring the impact of gender inequality on child development at the individual level.

**Open peer review.** To view the open peer review materials for this article, please visit http://doi.org/10.1017/gmh.2026.10139.

**Supplementary material.** The supplementary material for this article can be found at http://doi.org/10.1017/gmh.2026.10139.

**Data availability statement.** Data used are not openly available due to confidentiality of information warranted by the written informed consent. New projects or analyses can be discussed with the Federal University of Pelotas team.

**Author contribution.** NAC led the conceptualization, analysis and writing of the manuscript; CSG contributed to the study concept and design; LC assisted with the analysis; ABM and FW provided and curated the data; and all authors reviewed and approved the final version of the manuscript.

**Financial support.** Wellcome supported the 1993 Birth Cohort Study between 2004 and 2013 through grants 72403MA and 086974/Z/08/Z. Previous phases of the study were funded by the European Union, the Brazilian National Support Program for Centers of Excellence (PRONEX), the Brazilian National Research Council (CNPq) and the Brazilian Ministry of Health. NAC is supported by Centro de Interés Nacional IINARA, CIN250068, Agencia Nacional de Investigación y Desarrollo ANID Chile.

**Competing interests.** The authors report no conflict of interest related to this work.

**Ethics statement.** The Pelotas 1993 Birth Cohort study was approved by the Research Ethics Board of the Medical School of Federal University of Pelotas (protocols 029/2003, 158/2007, 05/2011). Informed consent was obtained from the cohort participants or by their parents when individuals were younger than 18 years.

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
