## [Reviewer Report]

This manuscript provides a novel and policy-relevant analysis of how couple-level gender inequality associates with adolescents’ developmental and mental health outcomes, leveraging data from Brazil’s long-standing Pelotas 1993 birth cohort. The findings suggest that higher gender equality within couples is positively associated with children’s educational attainment and quality of life, and possibly protective against depression, regardless of gender.

Overall, I find the paper interesting and innovative. However, I am concerned with several methodological issues.

My first concern is the inclusion of reproductive-health indicators in the CGII. While such items belong in population-level measures like the UN Gender Inequality Index, they are problematic at the couple level. Indicators such as teen childbirth and inadequate prenatal care may signal health risk and socioeconomic disadvantages instead of gender inequality. Because teen births and limited prenatal care are strongly associated with low birth weight, pre-term delivery, and later developmental problems, their inclusion could conflate household gender dynamics with maternal health or family SES. The authors could either (1) provide a clear theoretical rationale for treating maternal age at first birth and antenatal-visit adequacy as indicators of intra-couple bargaining power or (2) remove these items and report whether the main results hold. They can also analyze the reproductive and socioeconomic components separately to show which domain drives the associations.

Second, for the CGII itself, please report Cronbach’s alpha to demonstrate its internal consistency and support the index’s validity

Third, though the authors admit potential selection bias at the end, nearly 50 % of the cohort is missing CGII data, which is not trivial. The authors could consider comparing participants with complete CGII measures to those without using key SES and demographic variables, and/or apply inverse-probability weighting or similar techniques to test the robustness of their findings to potential selection bias.

Next, several questions arise about the modelling choices. Dividing the CGII into quartiles discards information and imposes arbitrary thresholds, the authors could consider capturing the non-linear effects directly by including an interaction term (squared CGII). Next, the decision to cap the mother-to-father ratios at 1.0 may obscure households in which women hold greater power, a potentially informative contrast for the research question. Lastly, key child-level confounders are missing. Low birthweight, for example, predicts later developmental outcomes and could influence couple-level inequality through reduced maternal earnings. If such variables are available, they should be included. If not, their absence needs to be acknowledged explicitly in the limitations.

Last, a minor point, the manuscript labels sequentially adjusted models “hierarchical.” Because “hierarchical models” usually denotes multilevel (mixed effects) techniques, this wording could mislead. Replace with other terms such as “stepwise” to avoid confusion.

---

## [Reviewer Report]

Referee report for “Parental gender inequality and their children’s educational attainment, quality of life, and mental health: an analysis from the Pelotas 1993 birth cohort in Brazil.

Summary

This study examined whether greater gender equality between parents is linked to improved outcomes for their children using data from the 1993 Pelotas Birth Cohort in Brazil. A novel Couple’s Gender Inequality Index (CGII), based on maternal vs. paternal differences in health, education, and income, was created to measure intra-household gender dynamics. Findings showed that children from more gender-equal families had significantly more years of education and better quality of life by age 18. While higher gender equality was also associated with lower risk of depression and emotional problems, these mental health associations were weaker after adjusting for confounders. Importantly, both boys and girls appeared to benefit from greater parental gender equality, especially in low-income families.

Overall View

This paper addresses an important and underexplored topic, how gender equality within the parental couple influences children’s long-term education, well-being, and mental health. By adapting the UN’s Gender Inequality Index to the family level and applying it to a large, longitudinal birth cohort in Brazil, the study makes a novel and meaningful contribution to global mental health and development research. The findings are broadly relevant for policy and interventions aiming to reduce intergenerational disadvantage, especially in low- and middle-income countries. However, before publication, the manuscript requires revisions to improve clarity, reporting transparency (especially around the CGII construction and statistical models), and consistency in interpreting results. The observed associations are compelling, but care is needed in how causal language is used, how interactions are presented, and how limitations are discussed.

Comments

Introduction

1. The introduction would flow better beginning with broader gender inequality concerns and then narrowing down to the gap your study fills.

2. Early on, make it clear that a higher CGII score indicates greater gender equality as this will help readers interpret the findings more easily later.

3. Phrases like “We here analysed…” should be improved. Use more standard academic phrasing such as “In this study, we analysed…”.

4. To help emphasize the contribution your study is making, highlight the originality of your family-level adaptation of the Gender Inequality Index.

Methods

5. The CGII construction is central to your study, so offering a bit more detail on how the components were scored and combined would make this clearer to readers trying to replicate or understand your work. While the text and Table 1 outline the components, a more explicit description in prose would be helpful. You should clearly state how the three dimensions were combined into a single index value. For example, did you create normalized sub-scores for each and then average them? Or did you use an approach analogous to the UN GII formula? Currently, you note: “Worse reproductive health and lower educational level and earnings from the mother compared to the father were considered to signal a more gender unequal relationship.” This explains the direction of each component but not the aggregation. Consider adding a sentence such as: “Each dimension was scored on a 0–1 scale (with 0 indicating maximum inequality and 1 indicating parity), and the CGII was calculated as the average of the three dimension scores for each couple.” (Please adjust this description to the actual method you used. If it was a sum or another formula, describe that accordingly.)

6. It would be useful to explicitly mention how participants were included, why some were excluded and how you handled cases where parent data was incomplete (e.g., non-cohabiting parents or missing income data). For instance, if a father’s income was missing, was that family excluded or was income imputed? It’s important to clarify, since non-cohabiting fathers or missing paternal info likely contributed to the reduced sample. You do include “living with both parents” as a confounder, implying some single-parent families might be included; clarify whether CGII was only defined for cohabiting couples or if data on non-resident fathers were used. This transparency will help readers trust more the CGII measure.

7. You mention your four outcomes, but readers may appreciate a few more details, like who completed the SDQ, the scoring range for QoL, or whether education was self-reported. In particular, for each outcome:

a) Educational Attainment: State how this was measured (e.g., “number of years of formal education completed by age 18 (self-reported at the 18-year visit)”. Indicate the range if known (e.g., 0–? years, and whether 12 years corresponds to high school completion in Brazil). This gives context that 8.9 years on average means many did not finish secondary school by 18, an important detail possibly worth noting in Discussion.

b) Quality of Life: You mention using the WHO Quality of Life instrument. Specify which version and whether you used an overall score or a particular domain. If it’s a domain or a transformed score, clarify that as well. Also, confirm that higher scores mean better QoL.

c) Depression at 18: You note a diagnosis via the Mini International Neuropsychiatric Interview (MINI). Clarify that this outcome is binary (depressed vs. not) at that age.

d) Emotional/Mental Health at 15: It appears you used the Strengths and Difficulties Questionnaire (SDQ) emotional problems subscale. Clarify this as: “emotional symptoms at age 15, measured by the Strengths and Difficulties Questionnaire (SDQ) emotional subscale”. Indicate the reporter (was it adolescent self-report at 15 or mother report?) and the score range. You might also mention why this age 15 measure was included (presumably to capture emerging emotional problems in mid-adolescence before the diagnostic assessment at 18). If the SDQ emotional score was treated as continuous, note that, or if you dichotomised “elevated symptoms” etc., specify the cut-off.

8. Clarifying a bit more the choice of models and how you interpreted the coefficients, would strengthen the Methods section. For instance: “We applied regression models appropriate to each outcome’s distribution. For count outcomes (e.g., years of education and SDQ emotional symptoms score), we used Poisson regression with robust standard errors to estimate relative differences (interpreted as percentage differences). For the binary depression outcome, we used logistic regression to estimate odds ratios (OR). For quality of life (approximately continuous), linear regression was used.” Ensure the text reflects exactly what was done. If you in fact used Poisson for the binary outcome to obtain risk ratios (a valid approach), state that instead (e.g., “Poisson regression was also used for the binary depression outcome to estimate relative risk, given the outcome frequency”). This level of detail is important for readers to understand how to interpret the reported “20.7% more years” (clearly a Poisson relative count result) versus an “X% of the odds” for depression (a logistic OR result).

Results

9. Your description of the sample is helpful, but readers might benefit from seeing the key characteristics in a table or in slightly more structured form in the text. For example, a table could show the distribution of maternal perinatal care, adolescent pregnancy, education levels, and income in the sample, possibly stratified by CGII quartile if relevant. Since you mention Figure 1 displays the distribution of CGII and its components, the reader gets a visual, but a table would provide precise numbers. If journal space is a concern, think about which format (text vs table) most clearly conveys this information. At minimum, ensure the text is complete: currently, you give percentages for extreme categories of perinatal visits (>=8 and <4) but not the middle category, consider mentioning the remainder (around ~35.8% had 4–7 visits, presumably). Similarly, you state 62.9% of parents had equal or greater maternal education than paternal, implying 37.1% of mothers had less education than fathers, that’s an important baseline showing inequality in education. You might state it explicitly for emphasis. And only 4.9% of mothers earned as much or more than fathers, meaning 95.1% of fathers were the primary earners, a striking figure to highlight. These details set the stage for why CGII variation exists.

10. You note the 2,852 analysed were 69.5% of those followed at 18, implying some attrition bias. It might be worth stating whether those included had any notable differences from those not included (for instance, were included families slightly higher socio-economic status or more likely to be intact? You might have checked this). If you have that info, a sentence could be added, e.g., “Compared to the full cohort, the subset with complete data had slightly higher family income and education, suggesting our analytic sample may be somewhat socioeconomically advantaged; thus results should be interpreted accordingly.” This can also be noted in Discussion limitations if not here. It’s also worth summarizing better the CGII components to give readers a sense of what inequality looked like in the cohort, like how common it was for mothers to earn less or be adolescent mothers.

11. Where you report interactions (e.g., between CGII and sex), an explanation not just that an interaction exists, but what it means (did one group benefit more? If so, how?) would benefit the section.

12. Adding a specific example for the QoL and depression outcomes, like a percentage or odds ratio, would help readers understand better the magnitude of associations.

13. The interaction with income is really interesting, but the explanation could be a bit clearer. Perhaps rephrase to show that low-income families seemed to gain more from equality.

Discussion

14. Acknowledge that your design is observational. It’s good that you avoid strong causal language generally, but terms like “impact” and “effect” were used in a causal sense in a few places (abstract conclusion, discussion opening). In the discussion, you should explicitly note that association does not prove causation. For example: “Although we refer to ‘effects’, these results are associative. It is possible that unmeasured factors contribute to both more egalitarian parent relationships and better child outcomes. For instance, parents with higher mutual respect might also possess other positive parenting qualities or socio-economic advantages that benefit children.” You did adjust for several confounders to mitigate this, which you can state: “We adjusted for baseline income, parental cohabitation, and other factors to approximate causal effects, but residual confounding (e.g., parental mental health or personality traits) could still explain some associations.”

15. It’s good that you had a specific hypothesis about girls benefiting more. Just make sure you clearly describe how your results did or didn’t support that expectation.

16. Consider suggesting why greater gender equality might be beneficial. What mechanisms might explain the associations?

17. The fact that low-income families seemed to benefit more is a compelling finding. You might want to highlight this implication for targeting interventions.

18. When listing your limitations, be sure to discuss potential biases from attrition and the fact that some elements of your index are static or not directly modifiable.

19. A closing paragraph that summarizes what your findings add to the field and what could be done next would really round out the discussion nicely.

Additional general comments

20. Go through the manuscript for minor writing style improvements. Avoid words like “firstly” and stick to past tense when describing your findings.

21. Also double-check small grammatical details. Phrases like “on the highest quartile” should be “in the highest quartile” to read more smoothly, and verify consistency: e.g., in one part you refer to “indices” available for 54%. It should likely be singular “index.”

22. In the abstract, where you state “The Couple’s Gender Inequality Index (CGII) was derived from maternal health, education, and income,” consider whether “maternal” should be replaced with “parental,” since the index appears to reflect characteristics of both members of the couple.

---

## [Reviewer Report]

Overall, the authors have responded thoughtfully to most of my comments. While I remain somewhat unconvinced by the explanation for not modeling non-linear effects of CGII using squared terms and interactions with gender, I recognize the authors’ rationale and note that concerns about interpretation are understandable. Non-linear interaction effects are common in the literature and can often be communicated clearly using tools such as marginal effects and predicted-value plots, and discretizing a continuous variable also entails functional-form assumptions, including stepwise effects. That said, I respect the authors’ modeling choice and appreciate their transparent discussion of this issue in the limitations section.